Report

# Autocrine Wingless constricts the *Drosophila* embryonic gut by Ca$^{+2}$-mediated repolarisation of mesoderm cells

Delia Ricolo [1,2,3✉], Francesca Tamba[1,2,3] & Jordi Casanova [1,3✉]

## Abstract

Wg/Wnt signalling—a highly conserved transduction pathway—has most commonly been found to be involved in patterning, cell fate, or cell proliferation, but less so in shaping organs or body parts. A remarkable case of the latter is the role of Wg signalling in the midgut of the *Drosophila* embryo. The *Drosophila* embryonic midgut is divided into four chambers that arise by the formation of three constrictions at distinct sites along the midgut. In particular, Wg is responsible for the middle constriction, a role first described more than 30 years ago. However, while some partial data have been obtained regarding the formation of this gut constriction, an overall picture of the process is lacking. Here we unveil that Wg signalling leads to this constriction by inducing *ClC-a* transcription in a subset of mesodermal cells. *ClC-a*, encodes a chloride channel, which in turn prompts a Ca$^{+2}$ pulse in these cells. Consequently, the mesoderm cells, which already showed some polarity, repolarise and in so doing so they reshape the microtubule organisation, therefore inducing the constriction of the cells.

Keywords Cell Polarity; Constriction; Endoderm; Mesoderm; Wingless
Subject Categories Cell Adhesion, Polarity & Cytoskeleton; Development; Signal Transduction

## Introduction

The midgut of the *Drosophila* embryo is formed by an endodermal layer surrounded by another layer of cells belonging to the visceral mesoderm (Poulson, 1950; Campos-Ortega and Hartenstein, 1985 and Fig. 1A,B). Upon development, the gut is divided into different chambers by three constrictions along its length (Fig. 1B). In particular, Wg signalling was found responsible for the formation of the middle constriction (Immerglück et al, 1990 and Fig. 1C), a process that has been studied since then (Reuter and Scott, 1990; Mathies et al, 1994; Mitchell et al, 2022). Indeed, *wg* is expressed in few cells of the visceral mesoderm at the place where the middle

constriction will form (Immerglück et al, 1990 and Fig. 1D,E). While it was initially suggested that Wg from the visceral mesoderm induces the constriction of the endoderm cells (Immerglück et al, 1990), it was later proposed that changes in the *wg*-positive cells of the visceral mesoderm mechanically impose the constriction on the underlying endoderm (Reuter and Scott, 1990), although this is still a matter of debate (Mitchell et al, 2022).

## Results and discussion

To precisely define the cells responsible for the Wg-induced constriction of the midgut, we specifically impaired the Wg pathway in *wg*-positive cells of the visceral mesoderm. To this end, and by means of the GAL4/UAS system (Brand and Perrimon, 1993), we first used a mesoderm-GAL4 driver (24B) to force the expression in the mesoderm layer of a construct coding for Axin (mat&met), a negative regulator of the Wg pathway (Hamada et al, 1999). We detected a failure in the formation of the middle midgut constriction (Fig. 1F). Given that Axin interferes with intracellular events of the Wg transduction pathway, these results identified the cells of the visceral mesoderm as the effectors of the Wg-induced constriction. Our observations are consistent with recent findings indicating that contractions of visceral muscle shape midgut constrictions (Mitchell et al, 2022).

We then used a *wg*-GAL4 driver to force the expression, specifically in the *wg*-expressing cells, of a construct coding for a dominant-negative form of dTCF (mat&met), the transcription factor that regulates gene expression in response to Wg signalling (van de Wetering et al, 1997; Brunner et al, 1997). In this case, we also detected a failure in the formation of the middle midgut constriction (Fig. 1G). As dTCF$^{DN}$ interferes with gene transcription downstream of the Wg transduction pathway, this result identified the *wg*-positive cells of the visceral mesoderm as the effectors of the Wg-induced constriction in an autocrine manner. Finally, the results with dTCF$^{DN}$ indicated that the Wg-induced constriction is a transcription-mediated response to Wg signalling (see below).

We sought to determine whether Wg signalling was sufficient to trigger gut constriction. To this end, we ectopically expressed *wg* throughout the visceral mesoderm by means of the 24B driver

[1]Institut de Biologia Molecular de Barcelona (CSIC), Barcelona, Catalonia, Spain. [2]Universitat de Barcelona, Barcelona, Catalonia, Spain. [3]Institut de Recerca Biomèdica de Barcelona (IRB Barcelona), The Barcelona Institute of Science and Technology (BIST), Barcelona, Catalonia, Spain. ✉E-mail: delia.ricolo@irbbarcelona.org; jcrbmc@ibmb.csic.es

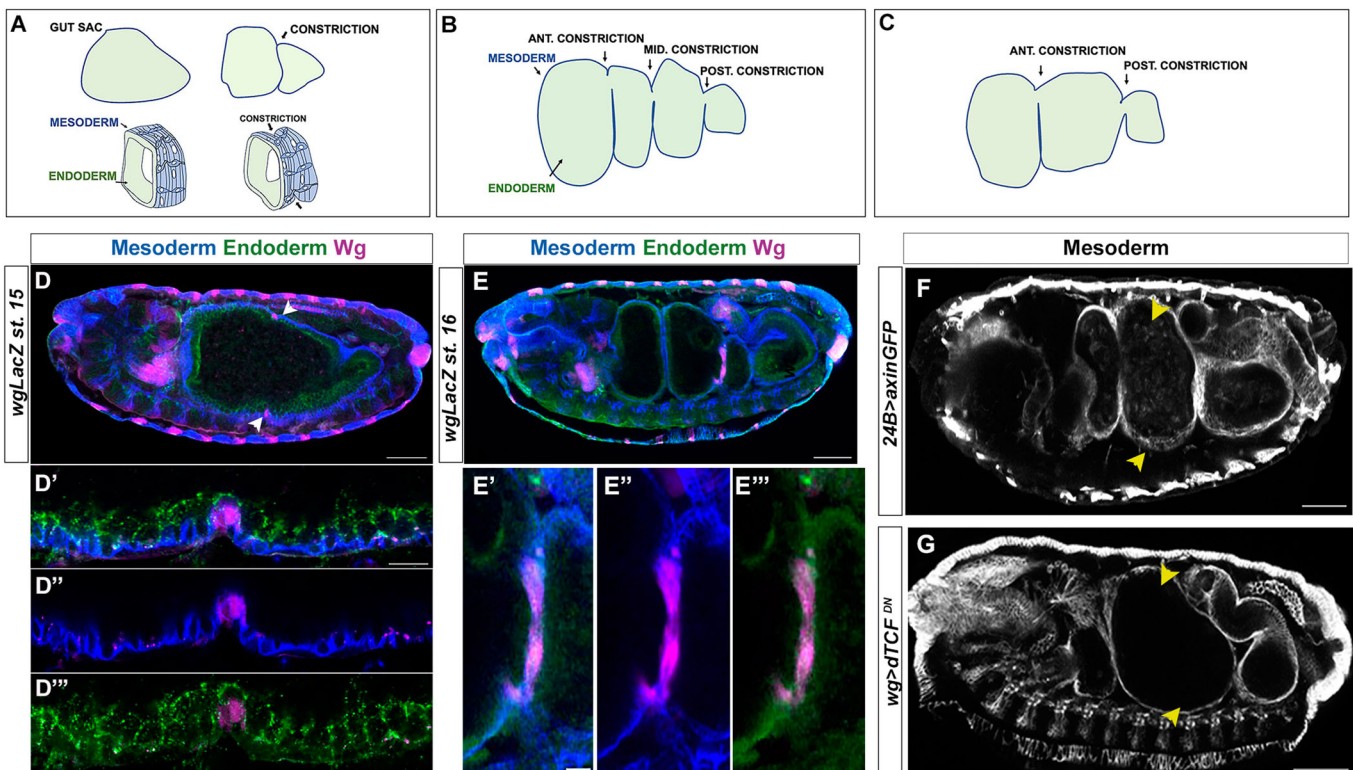

**Figure 1. The wingless signalling pathway in the visceral mesoderm is necessary for the middle midgut constriction.**

(A–C) Schematic representation of the midgut of *Drosophila* embryo (anterior is left, dorsal is top). At stage 15th, the gut sac undergoes the first constriction event, which constitutes the middle constriction at later stages (A). At stage 16th, three constrictions divide the gut in four chambers (B). Schematic illustration of a *wingless* mutant midgut, characterized by the absence of the middle constriction (C). The surrounding visceral mesoderm (in blue) pushes the inner endodermal layer (in green). (D, E) *wg*LacZ embryos at stage 15th (D) and 16th (E) stained with FasIII to recognise the visceral mesoderm (blue), De-caderin to detect the endoderm (green) and βgal to label wingless positive cells (magenta). Anterior side is on the left, dorsal side is top, scale bar 50 µm. (D', E''') magnifications to show the *wg* positive cells (magenta) in the visceral mesoderm (blue) contributing to the middle constriction, Scale bar 10 µm. (F, G) Overexpression of Axin under the control of the mesodermal driver 24B (F) and overexpression of a dominant negative form of dTCF in the *wg* domain (G), result in a failure in the formation of the middle constriction (arrowheads), stained with FasIII to visualize the visceral mesoderm. Anterior is left, dorsal is top, scale bar 50 µm.

(mat&met). Visceral mesoderm migration was disrupted, thereby also affecting the endoderm and thus impeding the correct formation of the gut (Fig. EV1). However, in these circumstances we found the overall visceral mesoderm to display some of the features associated with the middle constriction (see below).

The above results indicate that, to examine the cellular bases of the midgut constriction, we should focus on the contracting cells of the visceral mesoderm. Previous results from electronic microscopy have revealed in the constricting cells a transient and dense accumulation of microtubule bundles oriented in the direction of the constriction, a microtubule arrangement that coincides with the time of constriction as it is not detected at earlier or later times (see Fig. 8 in Reuter and Scott, 1990). These observations point to an active role for microtubules in driving constriction or, alternatively, the microtubule organisation as a consequence of the cell changes associated with the constriction event.

To examine whether the microtubules participate in driving constriction, we used different conditions to impair microtubule polymerisation and function. On the one hand, we induced the overexpression of *spastin* (*spas*), a gene coding for a microtubule severing enzyme (Roll-Mecak and Vale, 2005), which is efficiently used to disassemble microtubules (Sherwood et al, 2004). Upon

*spas* overexpression in the *wg*-expressing cells with the *wg*-GAL4 driver, the middle midgut constriction failed to form (Fig. 2B) conversely to the wt (Fig. 2A). Of note, upon *spas* overexpression throughout the visceral mesoderm with the 24B driver, other midgut constrictions also failed to form (Fig. 2C). These observations suggest that while the anterior and posterior constrictions are not under the control of Wg signalling, they both seem to be induced by a similar microtubule-based mechanism (see below). To gain an independent assessment of role of microtubule polarisation in midgut constriction, we resorted to *Eb1*, a gene encoding a protein that binds to the ends of growing microtubules (Mimori-Kiyosue et al, 2000; Rogers et al, 2002) and whose dominant negative form interferes with microtubule growth, thereby impairing their orientation (Bulgakova et al, 2013). *Eb1*[DN] overexpression (mat&met) with the 24B driver yielded similar results to those obtained by *spas* overexpression (Fig. 2D).

We then explored the mechanisms that might account for microtubule polarisation. Special attention was given to the fact that the microtubule network of the follicle cells of the *Drosophila* ovary becomes polarised upon the activity of Par-1 kinase (Doerflinger et al, 2003), which plays a conserved role in cell polarity from *C. elegans* to mammals. While Par-1 has been found

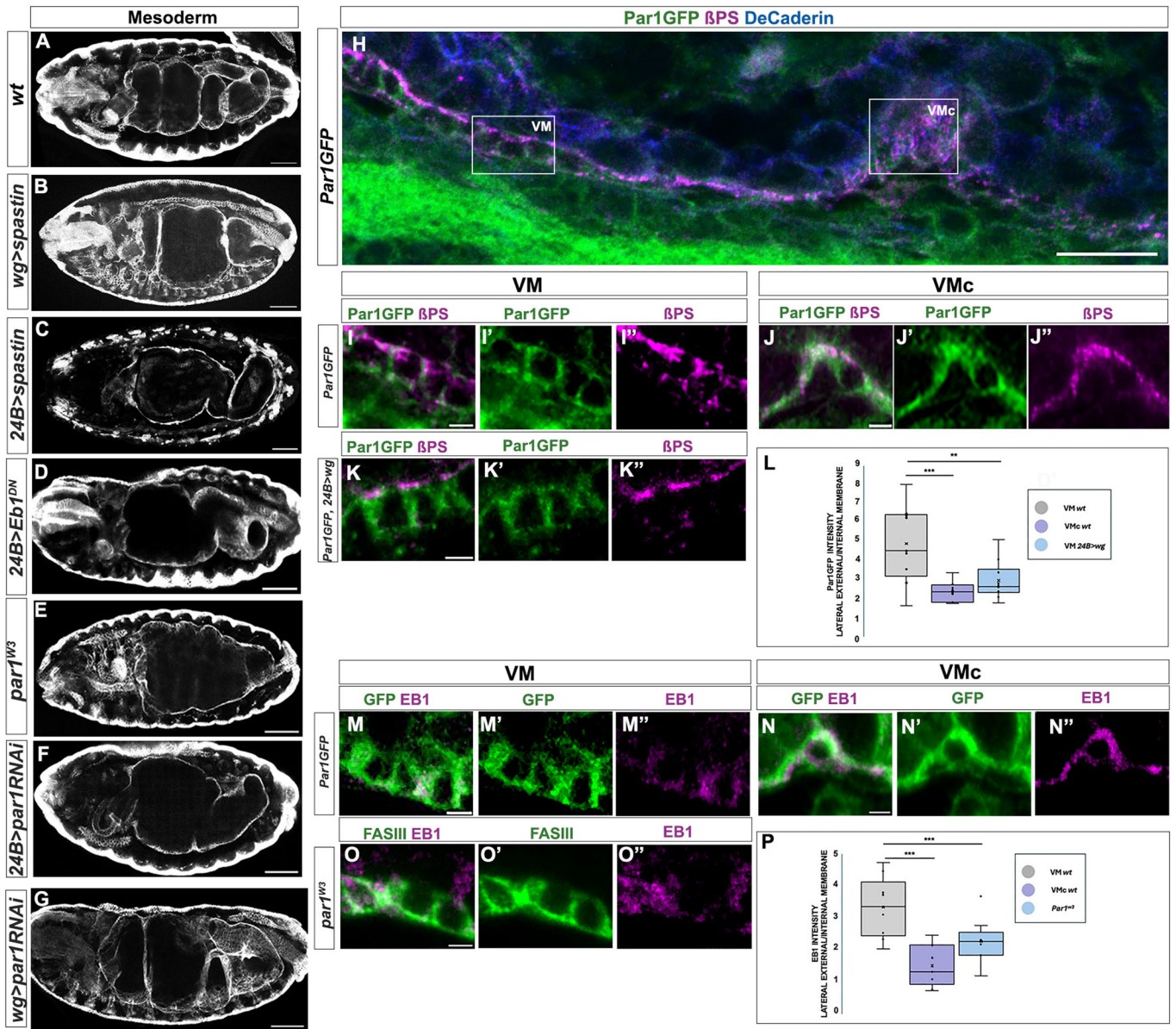

**Figure 2. Visceral mesoderm polarity in non-constricting and in the middle constriction cells.**

(A–G) Stage 16th wild-type (A), wg>spastin (B), 24B>spastin (C), 24B>eb1DN (D), par1w3 (E), 24B> par1RNAi (F) and wg>par1RNAi (G) embryos stained with FasIII to visualise the mesoderm. Note that the embryos in (B) and (G) are the only ones in which we used a driver acting at the middle constriction, and thus only the middle constriction is affected, while (C), (D) and (F) a mesoderm driver is used, thus affecting the other constrictions as well. Anterior side is on the left and dorsal is up, scale bars 10 µm. (H) Detail view of the midgut from a par1::GFP embryo stained with antibodies against βPS Integrin (magenta), to visualize the visceral mesoderm – endoderm interface and against GFP (in green) to visualize Par1. VM: visceral mesoderm; VMc: visceral mesoderm at middle constriction, scale bar 2 µm. (I, J) Close-up details of a gut region non-constricting (I, VM square in (H)) and a gut region from the middle constriction (J, VMc square in (H)). In the non-constricting area, Par1 accumulates less at the internal membrane, facing the endoderm, where βPS integrin accumulates, while at the middle constriction Par1 accumulates strongly at the internal membrane overlapping with βPS Integrin, scale bar 2 µm. (K) Detail of the visceral mesoderm of Par1GFP, 24B>wg embryos. Par1 is seen less asymmetrically distributed than in the wild-type, scale bar 2 µm. (L) Quantification of Par1GFP distribution as the ratio between the signal intensity at the lateral-external side and the internal side facing the endodermal tissue, at the non-constricting visceral mesoderm (grey), at the middle constriction (purple) at the visceral mesoderm of 24B>wg embryos (light blue) (n = 9 cells from 3 different embryos for each condition) Mann–Withney Test was used for comparison between pairs. P values are 0.01 (***), 0.05(**). Box plot VM wt: Min = 1.63, Q1 = 3.18, Median = 4.48. Q3 = 6.35, Max = 7.92. Box plot VMc wt: Min = 1.75, Q1 = 1.85, Median = 2.46, Q3 = 2.7. Max = 3.34. Box plot VM 24B>wg: Min = 1.79, Q1 = 2.42, Median = 2.7, Q3 = 3.37. Max = 5.07. (M, N) Detail of the visceral mesoderm in non-constricting (M) and in the middle constriction cells (N) of par1GFP embryos stained with an EB1 antibody to show microtubule polarisation. EB1 signal strongly correlates with Par1 accumulation. Scale bar: 2 µm. (O) Detail of the visceral mesoderm from a par1w3 mutant embryo stained with FasIII and EB1 antibodies. EB1 signal is much disorganised than in wild-type embryos. In (H–O), the internal side of the the visceral mesoderm is up, anterior is left and scale bars correspond to 2 µm. (P) Quantification of EB1 distribution as the ratio between the signal intensity of the lateral-external side and the internal side facing the endodermal tissue, at the non-constricting visceral mesoderm (pink), at the 2nd constriction (purple) embryos and at the visceral mesoderm of par1w3 embryos (light blue) (n = 9 cells from 3 different embryos for each condition). Mann–Withney Test was used for comparison between pairs. P value 0.01 (***). Box plot VM wt: Min = 1.98, Q1 = 2.38, Median = 3.31, Q3 = 3.72. Max 4.70, Box plot VMc wt: Min = 0.67, Q1 = 0.86, Median = 1.26, Q3 = 1.89. Max = 2.40. Box plot VM Parw3: Min = 1.13, Q1 = 1.79, Median = 2.22, Q3 = 2.49 Max = 3.64.

to be associated mainly with epithelial cell polarity, it is also active in other cell types, such as the *Drosophila* oocyte (for a review see Lang and Munro, 2017).

First, using an endogenous GFP tag version of Par-1, we confirmed its presence in the visceral mesoderm. In fact, in the visceral mesoderm, we found a non-uniform cortical distribution of Par-1 along the cell membranes, a general feature of this kinase. To better characterise Par-1 distribution, we studied at the same time the distribution of the βPS integrin subunit, which more strongly accumulates at the membrane of the visceral mesoderm cells that face those of the endoderm (Fig. 2H,I). In particular, we found Par-1 cortical accumulation along the membrane at places of no or lower accumulation of βPS (Fig. 2H,I,L). To discard that the uneven Par-1 distribution might reflect an uneven accumulation of the cytoplasm due to a particular cell shape, we also studied at the same time the pattern detected by the KDEL antibody in the ER (see Methods), thus confirming a distinct subcellular localisation of the Par-1 cortical signal (Fig. EV2). Interestingly, we observed a different pattern of Par-1 accumulation in the constricting *wg*-positive cells of the visceral mesoderm as these cells showed higher coincident localization of cortical Par-1 and membrane βPS facing the endoderm (Fig. 2H,J,L). Indeed, Par-1 distribution in the whole visceral mesoderm was modified upon ectopic expression of *wg*. Although the ectopic expression of *wg* along the visceral mesoderm prevented its normal migration (see above), in this case we detected an enhancement of cortical Par-1 accumulation along the cell membrane facing the endodermal cells in all the cells of the visceral mesoderm, in the line of the distribution in the cells of the second constriction in wild-type embryos (Fig. 2K,L).

Furthermore, and consistent with the role of Par-1 in microtubule polarisation in other tissues (e.g., Doerflinger et al, 2003), and with the above established role of microtubules in the middle gut constriction, the middle constriction in *par-1* mutant embryos failed to develop (Fig. 2E). The same phenotype was found upon *par-1*- downregulation in the visceral mesoderm by means of a UAS-*par-1* RNAi construct under the control of the mesodermal 24B driver (Fig. 2F). Similarly, specific downregulation by means of the same UAS-*par-1* RNAi just in the *wg* domain also impaired middle constriction formation (Fig. 2G), pointing to a precise requirement in the constricting cells.

Again, like *spas* general expression, *par-1* downregulation in the overall visceral mesoderm not only impaired the establishment of the middle constriction but also that of the other midgut constrictions. These observations reinforce the notion that while the anterior and posterior constrictions are not under the control of Wg signalling, they are induced by a similar polarity-based mechanism.

We next examined whether the uneven distribution of Par-1 in the visceral mesoderm did indeed correlate with microtubule polarisation in these cells. To this end, we used an antibody against the EB1 protein as a marker of the microtubule polarity. Whether there is some controversy about how fixation may affect EB1 accumulation at microtubules, our fixation protocol (see Methods) has proven reliable to give a snapshot of EB1 accumulation, and thus microtuble polarization, at the time of fixation (see the discussion on the peer review section of Ricolo and Araujo, 2020). In fact, as for Par-1, we also found that EB1 accumulated preferentially away from the internal membrane of the mesoderm cells (not facing the endoderm) (Fig. 2M,P) while in constricting

cells it accumulated preferentially at the interior membrane (facing the endoderm) (Fig. 2N,P). Thus, the polarisation of the microtubules on the visceral mesoderm cells coincided with the uneven accumulation of Par-1. Moreover, we observed a more disorganised distribution of EB1 in *par-1* mutants (Fig. 2O,P), thereby pointing to Par-1 being necessary for the correct polarisation of the microtubules.

Recent results have shown that the asymmetric distribution of Par-1 in the oocyte depends on the combined activity of the non-muscle myosin II (MyoII) and its chaperone Unc-45 (Doerflinger et al, 2022). Consistent with a similar role of Unc-45 in Par-1 function in the visceral mesoderm, the midgut constrictions of *unc-45* mutants also failed to form (Fig. EV3). In the oocyte, Par-1 function is associated with a strong enrichment of the di-phosphorylated form of the MyoII regulatory light chain (MRLC), encoded in *Drosophila* by the *spaghetti squash* (*sqh*) gene. Interestingly, in the course of this work, the results from Streichan and collaborators also showed a critical role for Myosin in the gut constriction as the constriction was impaired upon RNAi-mediated downregulation of the calcium mediated kinase that phosphorylates the MyoII light chain; in another experiment they also showed the gut constriction to be dependent on calcium dynamics. (Mitchell et al, 2022). We thus decided to examine what might trigger a calcium flux in the visceral mesoderm.

As indicated above, the middle constriction of the midgut is dependent on Wg activity in a subset of cells in the visceral mesoderm in an autocrine manner. As we also found that the role of Wg in the constriction is mediated by activation of gene transcription, we sought to identify the transcriptional targets of Wg that might mediate constriction induction. The gene *teashirt* (*tsh*) might be such a target as: 1) it is expressed in the anterior and middle midgut constrictions; 2) it is required for their formation; and 3) *tsh* expression on the middle midgut constriction is dependent on Wg signalling (Mathies et al, 1994; for a picture of the phenotype see Fig. 3F compared to the wt in Fig. 3E). Thus, *tsh* is a candidate for triggering—directly or indirectly—the calcium flux in the cells of the middle constriction of the visceral mesoderm.

*tsh* codes for a transcription factor and is expressed in many other organs and tissues, including the Malpighian tubules—the *Drosophila* renal tubules—where it regulates the expression of *ClC-a*, a gene encoding a chloride channel (Delhom et al, 2013). Interestingly, *ClC-a* is also expressed in defined regions of the midgut (BDGP) and the depolarisation associated with the activity of *CLCN2*—the *ClC-a* human ortholog—has been related to an increase in cytosolic $Ca^{+2}$, probably due to the opening of voltage-gated calcium channels (Fernandes-Rosa et al, 2018).

To check whether *ClC-a* expression in the midgut participates in establising the middle constriction, we first analysed its expression in the visceral mesoderm in more detail. Using a line harbouring a GAL4 insertion in the *ClC-a* gene, we identified discrete expression in a subset of cells of the visceral mesoderm just before the initiation of the middle constriction (Fig. 3A). Moreover, by means of an anti-Wg antibody, we found that this subset coincided with the *wg*-positive cells leading the constriction (Fig. 3B). Finally, as development proceeded, the constriction was clearly linked to *ClC-a* expression (Fig. 3C). Furthermore, by means of the same *ClC-a*-GAL4 line, we observed that *ClC-a* expression in the middle midgut constriction was abolished in a *tsh* mutant background (see Methods for genotypes Fig. 3D). Interestingly, *ClC-a* was also

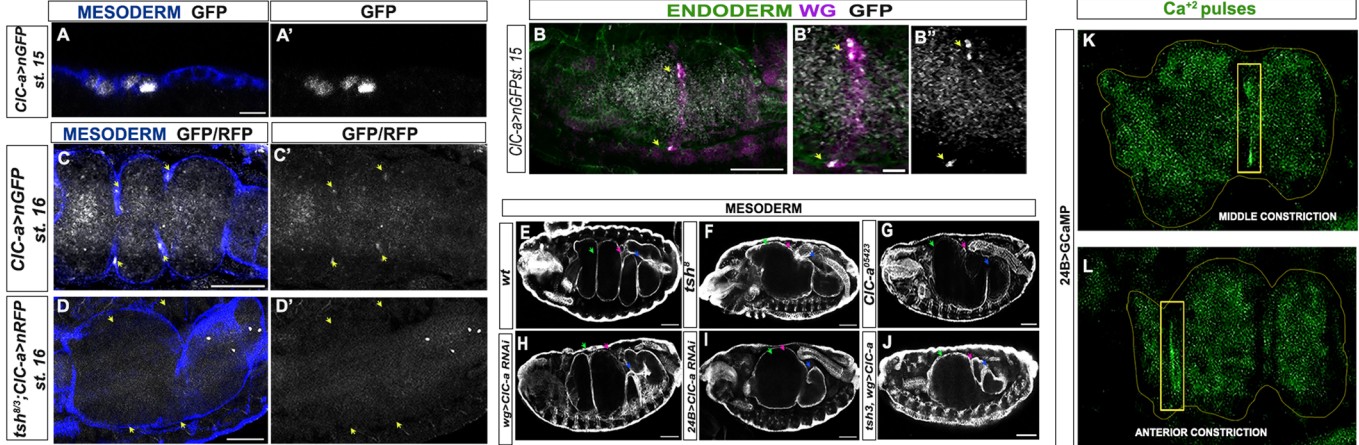

**Figure 3. The Wg pathway triggers Ca$^{+2}$ flux in the visceral mesoderm via *ClC-a* regulation.**

(A–D) *ClC-a* > nGFP embryo at stage 15th. *ClC-a* is expressed before the constriction forms in few cells of the visceral mesoderm (A) coincident with *wg* expression (B). At stage 16th *ClC-a* positive cells correspond with the anterior and middle constriction (yellow arrows) (C). *tsh$^8$/tsh$^3$;ClC-a* > nRFP embryo at stage 16th (D). *ClC-a* is not detected at the visceral mesoderm. Antibodies against GFP and RFP to label *ClC-a* positive cells in grey, against FasIII to visualise the visceral mesoderm in blue and against Wg in magenta. Anterior side on the left and dorsal up. Scale bars 5 μm in A, 10 μm in B, and 50 μm in (B), (C) and (D). (E–J) Stage 16th wt (E), *tsh$^8$* (F), *ClC-a$^{05423}$*(G), *wg* > *ClC-a* RNAi (H), 24B>*ClC-a*RNAi (I) and *tsh$^3$;wg* > *ClC-a* (J) embryos stained with anti FasIII to recognize the visceral mesoderm. Scale bar 50 μm. (K, L) Frames from supl. Movie EV1. A 24B>GCamP embryo showing the Ca$^{2+}$ pulses associated with the middle (K) and anterior (L) constrictions (yellow squares).

expressed in the anterior midgut constriction, where *tsh* is also expressed, and *tsh* mutants showed no expression at this site (Fig. 3C,D). Thus, *ClC-a* expression in the anterior and middle midgut constrictions is directly or indirectly dependent on the Tsh transcription factor.

Thus, *ClC-a* is expressed at the right place and at the right time to have a putative role in the middle midgut constriction. Further proving a fundamental role for the ClC-a chloride channels in this process, both the anterior and the middle midgut constrictions, sites of *ClC-a* expression, were affected in *ClC-a* mutant embryos failed to form (see Methods) (Fig. 3G). Moreover, RNAi-mediated downregulation of ClC-a just in the wg domain was sufficient to impair the middle constriction pointing to the specific requirement of the ClC-a channels in the constricting cells (Fig. 3H). Again, both the anterior and the middle midgut constrictions were affected by RNAi-mediated downregulation of *ClC-a* along the entire visceral mesoderm using the 24B driver (Fig. 3I). Finally, while other downstream targets of tsh may contribute to the midgut constrictions it is worth noting tha forced expression of *ClC-a* in a *tsh* mutant background by means of a wg-GAL4 driver is able to partially rescue the formation of the middle constriction (Fig. 3J).

To further confirm the link between the ClC-a channels and the Ca$^{+2}$ pulse, we used the GCaMP6 construct, a genetically encoded fluorescent Ca$^{+2}$ indicator (see Methods), and monitored in vivo the fluorescence signal upon expression of the reporter in the visceral mesoderm. In an otherwise wild-type background we could clearly detect the signal in groups of cells in the visceral mesoderm (Movie EV1). One group coincided with the cells that undergo the middle constriction (Fig. 3K) while another group coincided with those cells that undergo the anterior (Fig. 3L). However, we did not detect these signals upon expression of the same reporter in the visceral mesoderm of *ClC-a* mutant embryos (Movie EV2). Furthermore, we found that knock-down of calcium signalling by means of a dominant negative form of the SERCA calcium pump,

which removes midgut folds (Mitchell et al, 2022) and Fig. 4B (see Fig. 4A for a wt), also inhibits the EB1 relocalisation triggered by wg signalling (Fig. 4C–E).

Overall, our results identify a set of critical elements to complete the path linking the Wg signal in the visceral mesoderm to the formation of the middle midgut constriction in the endoderm (Fig. 5). Together with previous results (Reuter and Scott, 1990; Mathies et al, 1994; Delhom et al, 2013; Mitchell et al, 2022), we can now establish the following process. First, Wg secreted from a group of cells in the visceral mesoderm activates in these same cells the expression of the gene coding for the Tsh transcription factor (Mathies et al, 1994). Tsh is then responsible for the activation of the gene encoding the ClC-a chloride channel in these cells. *ClC-a* needs not to be the only target of Tsh but our results show that these channels are coupled with an increase in intracellular calcium, which, through a calmodulin-mediated mechanism, activates the myosin light chain kinase (MLCK), which in turn activates MyoII through phosphorylation of its regulatory light chain (MRLC). MyoII, together with the unc-45 chaperone, is responsible for the relocalisation of Par-1 in these cells of the visceral mesoderm. Par-1 relocalisation is then instrumental in the polarisation of the microtubule network, which is required for the constriction of the cells of the visceral mesoderm (Fig. 4). This constriction leads to the constriction of the neighbouring endoderm and brings about the middle midgut constriction. This way, the Wg signal is transduced into the shaping of an organ, in this case the embryonic gut.

Three features of the constriction process deserve further comment. First, while previous experiments showed in the constricting cells a transient and dense accumulation of micro-tubule bundles oriented in the direction of the constriction (Reuter and Scott, 1990), we now show that microtubules have an active role in the constriction. Moreover, the particular orientation of the microtubules also appears to be crucial as interfering with the mechanisms responsible for microtubule orientation also impair

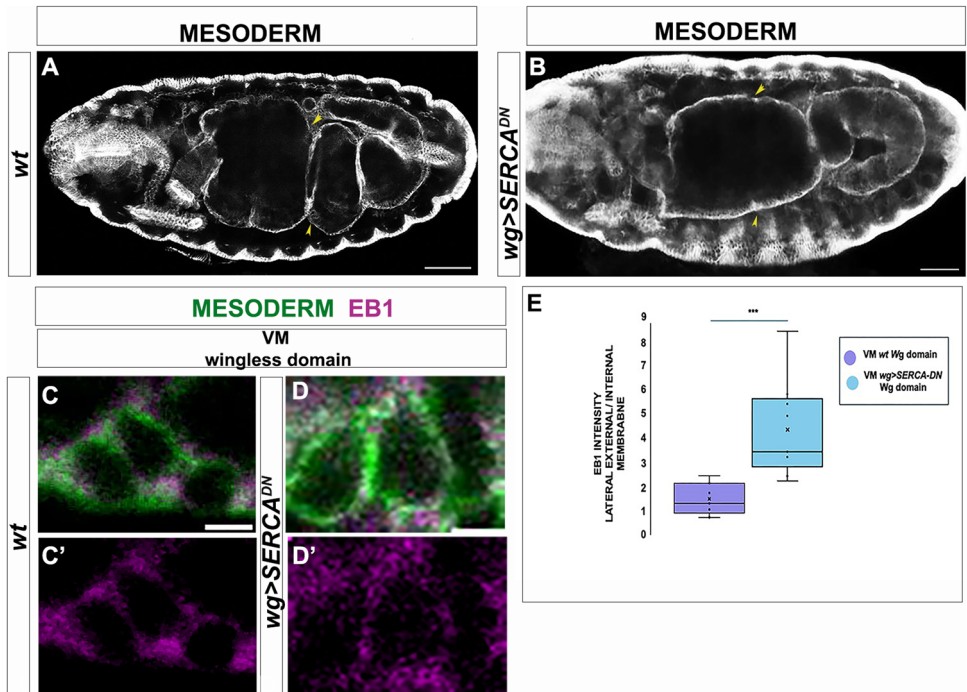

**Figure 4.  Knock-down of Ca²⁺ signalling impairs the change of MT polarity triggered by *wg* singalling.**

(A) *wt* embryo at early stage 16th, when the middle and the posterior constriction are formed (yellow arrows). (B) A *wg* > *SERCA*^DN embryo at early stage 16th. Note that the middle constriction is not formed (yellow arrows). Scale bar 50 μm. (C, D) Details of mesodermal cells control (C) and cells upon inhibition of calcium flow (D), stained with FasIII (in green) and EB1 (in magenta). Internal memebrane is up, external memebrane is down, scale bare 2 μm. EB1 accumulation is normal in control cells but not in *SERCA*^DN overexpressing cells. (E) quantification of EB1 intensity (*n* = 9 cells from 3 different embryos for each condition). Mann–Withney Test was used for comparison between pairs. *P* value 0.01 (***). Box plot VMc *wt*: Min = 0.67, Q1 = 0.86, Median = 1.26, Q3 = 1.89. Max = 2.40. Box plot VM *wg* domain: Min = 2.2, Q1 = 3.2, Median = 4.15, Q3 = 5.8. Max = 8.4.

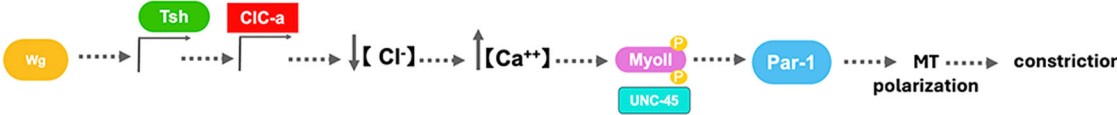

**Figure 5.  Schematic representation of the proposed role for Wg signalling in the formation of the middle midgut constriction.**

General scheme of the molecular mechanism; see text for details.

cell constriction. It is remarkable that the same elements contributing to microtubule orientation in epithelial tissues and in the oocyte are also at place in their organisation in the mesodermal cells (see below). How microtubules participate in the cell constriction? Do they contribute to some kind of actomyosin constriction? While this is an open possibility it has to be noted that a particular actin organisation has not been identified in these constricting cells, conversely to what has been reported for mirotubules (Reuter and Scott, 1990). In this regard it is also worth mentioning that recent work has drawn attention to the role of microtubule forces in cell reshaping (Singh et al, 2018; Takeda et al, 2018; Röper, 2020).

Second, different homeotic genes expressed in the visceral mesoderm are required for midgut constrictions (Bienz and Tremml, 1988; Reuter and Scott, 1990; Mathies et al, 1994; Mitchell et al, 2022); in the case of the middle constriction, all the effects of

these homeotic genes can be explained by their effect on establishing *wg* expression (Immerglück et al, 1990; Reuter and Scott, 1990; Mathies et al, 1994). Such mechanism could be instrumental in integrating the information from broadly expressed genes into a limited restricted domain. Likewise, other distinct combinations of homeotic genes and their downstream targets might explain the formation of the other two midgut constrictions, the anterior and the posterior. However, while these constrictions are Wg-independent, some Wg downstream elements reported herein also participate in their formation. Thus, for example, *ClC-a* expression induced by Tsh in a Wg-independent way in an anterior midgut domain also contributes to the anterior constriction. In another example, interference with microtubule polarisation not only impairs the middle midgut constriction but also the other two, thereby suggesting a common microtubule-mediated mechanism for the three constrictions. Given these observations, a scenario

emerges in which the three midgut constrictions are elicited by a common polarity-dependent mechanism triggered by somehow divergent paths. In this scenario, the recruitment of a common mechanism under the control of a new trigger would be sufficient to generate a new constriction, as is the case of Wg eliciting the middle one.

Finally, a third feature to be considered from our results is the relevance of polarity cues in the mesoderm. In fact, the epithelial state of the mesoderm monolayer, achieved at the end of gastrulation, has been previously underappreciated (as mentioned in Sun and Stathopoulos, 2018). While some asymmetric distribution of membrane proteins in the visceral mesodermal has already been shown, here we also show that this is the case for βPS at the membrane of mesodermal cells facing the endoderm. Moreover, we show how the overall polarity of the visceral mesoderm, and even more so the dynamic control of this polarity, is essential for morphogenesis. While it is a matter of debate whether the mesodermal polarity can be described as apicobasal polarity, we note that critical elements of the apicobasal polarity mechanism such as cortical Par-1 is also unevenly distributed in the visceral mesoderm cells. Moreover, the same mechanism involved in establishing the Par-1-mediated polarity in epithelial tissues is also at work in the visceral mesoderm. These results further extend the commonalities between the porperties of cells of different tissues or layer origins.

# Methods

### Reagents and tools table

| Reagent/resource | Reference or source | Identifier or catalog number |
|---|---|---|
| **Drsosphila melanogaster strains** | | |
| *wg*-LacZ | | BDSC 51661 |
| *wg*-GAL4 | | BDSC 83627 |
| 24b-Gal4 | | BDSC1767 |
| *ClC-a*-GAL4 | | BDSC 66801 |
| UAS-wg -HA | | BDSC 5918 |
| UAS-*axin*-GFP | | BDSC 7225 |
| UAS-*dTCF*$^{DN}$ | | BDSC 4784 |
| *UAS-spastin*-EGFP | D. A. Daga | |
| UAS-*ClC-a*-RNAi | | BDSC 53337 |
| *UAS-red stinger* | | BDSC 8547 |
| UAS-GCaMP6 | | BDSC 42748 |
| UAS-Eb1$^{DN}$-mCherry | | DBSC 58722 |
| UAS-*par-1*-RNAi | D. St Johnston | |
| *UAS-ClC-a* | D. M. Morey | |
| UAS- SERCA-DN | | DBSC58972 |
| *unc-45*$^{4F2-4}$ | D. St Johnston | |
| *par1-GFP* | D. St Johnston | |
| *par-1*$^{W3}$ | | BDSC 99513 |
| *tsh*$^8$ | | BDSC 50763 |
| *tsh*$^3$ | | BDSC 8883 |
| *ClC-a*$^{MI05423}$ | | BDSC 43680 |

| Reagent/resource | Reference or source | Identifier or catalog number |
|---|---|---|
| **Antibodies** | | |
| Rat anti-DE-cad | DSHB | DCAD2 |
| Mouse anti-Fas-III | DSHB | 7G10 |
| Mouse anti-β-PS | DSHB | CF6G11 |
| Mouse ant-Wingless | DSHB | 4D4 |
| Rabbit anti-Eb1 | Rogers et al, 2002 | |
| Goat anti-GFP | Abcam | ab3373/ab6673 |
| Chicken anti-β-gal | Abcam | |
| Rabbit anti-RFP | Abcam | ab62341 |
| Mouse anti-KDEL | Stressmarq Biosciences | SMC-539 |
| **Secondary Antibodies** | | |
| Donkey anti-mouse Alexa 555 | LIFE TECHNOLOGIES/ Thermofisher Scientific | A31570 |
| Donkey anti-mouse Alexa488 (Polyclonal) | LIFE TECHNOLOGIES/ Thermofisher Scientific | A-21202 |
| Donkey anti-mouse Alexa647 (Polyclonal) | LIFE TECHNOLOGIES/ Thermofisher Scientific | A31571 |
| Donkey anti-rabbit Alexa555 (Polyclonal) | LIFE TECHNOLOGIES/ Thermofisher Scientific | A31572 |
| Donkey anti-rabbit Alexa488 (Polyclonal) | LIFE TECHNOLOGIES/ Thermofisher Scientific | A-21206 |
| Donkey anti-rabbit Alexa647 (Polyclonal) | LIFE TECHNOLOGIES/ Thermofisher Scientific | A31573 |
| Donkey anti-goat Alexa 555 (Polyclonal) | LIFE TECHNOLOGIES/ Thermofisher Scientific | A21432 |
| Donkey anti-goat Alexa 488 (Polyclonal) | LIFE TECHNOLOGIES/ Thermofisher Scientific | A32814 |
| Donkey anti-rat Cy3 (Polyclonal) | Jackson ImmunoResearch | 712-165-150 |
| Goat anti-rat Alexa 488 (Polyclonal) | LIFE TECHNOLOGIES/ Thermofisher Scientific | A32814 |
| Goat anti-Rat Alexa Fluor647 (Polyclonal) | LIFE TECHNOLOGIES/ Thermofisher Scientific | A11006 |
| Goat anti-Mouse Cy3 (Polyclonal) | JACKSON INMUNORESEARCH | 115-165-003 |

## Fly strains and genetics

All fly stocks were raised at 25 °C on standard flour/agar *Drosophila* media. The GAL4/UAS system (Brand and Perrimon, 1993) was used at 29 °C to drive the expression of transgenes in particular regions of the embryo. The following strains were provided by the Bloomington Drosophila Stock Center: *wg-LacZ* (BDSC 51661), *wg-GAL4* (BDSC 83627), *UAS-wg-HA* (BDSC 5918), *UAS-axin-GFP* (BDSC 7225), UAS-*dTCF*$^{DN}$ (BDSC 4784), *tsh*$^8$ (BDSC 50763), *tsh*$^3$ (BDSC 8883), *ClC-a*$^{MI05423}$(BDSC 43680), UAS-*ClC-a*-RNAi (BDSC 53337), *ClC-a*-GAL4 (BDSC 66801), *UAS-ClC-a* (M. Morey, University of Barcelona)(Plazaola-Sasieta et al, 2019), *UAS-red stinger* (BDSC 8547), UAS-GCaMP6 (BDSC 42748), *24B-GAL4* (BDSC1767), *UAS-Eb1*$^{DN}$-mCherry (DBSC 58722), *UAS-SERCA-DN* (DBSC58972), *par-1*$^{W3}$ (BDSC 99513). The *par-1*-GFP, *unc-45*$^{4F2-4}$ and UAS-*par-1*-RNAi stocks were kindly provided by D. St

Johnston (Gurdon Institute, University of Cambridge) (Doerflinger et al, 2022) and the *UAS-spastin*-EGFP by A. Daga (University of Padova) (Trotta et al, 2004). A *yw* strain was used as a control.

## Immunohistochemistry, image acquisition and processing

Standard protocols for immunostaining were applied. All stage embryos, collected on agar plates overnight (O/N), were dechorionated with bleach and fixed for 20 min, or 10 min for microtubule staining, in 4% formaldehyde, PBS/Heptane 1:1. We have tested many different conditions in order to better detect both microtubules and other cell components within the whole embryo and the fixing protocol we use in this work gives us the best results for all structures we need to detect (see the peer review section of Ricolo and Araujo, 2020). Washes were done with PBT (PBS, 0.1% Tween). Primary antibody incubation was performed in fresh PBT-BSA 0.5% O/N at 4 °C. Secondary antibody incubation was done in PBT-BSA at room temperature (RT) in the dark for 2 h. The following antibodies were used: Rat anti-DE-cad (DCAD2, DSHB) 1:100; mouse anti-Fas-III (7G10, DSHB) 1:20; mouse anti-$\beta$-PS (CF.6G11, DSHB) 1:20; rabbit antiEb1 (Rogers et al, 2002) 1:100; goat anti-GFP 1:300 (Abcam), anti-RFP 1:200 (Abcam) chicken anti-$\beta$-gal (Abcam) 1:300, mouse antiKDEL (Stressmarq Biosciences) 1:200, Cyanin conjugated secondary antibodies (Jackson Immuno-Research) 1:300 and Alexa conjugated secondary antibodies (Thermofisher) 1:300. Confocal images of fixed embryos were obtained either with a Leica TCS-SPE or a Zeiss 780. Images were processed using Fiji and assembled using Photoshop.

## Time lapse imaging

Dechorionated embryos were immobilized with membrane on a coverslip and covered with Oil 10 s Voltalef (VWR). To visualize in vivo the $Ca^{+2}$ pulses in the gut, the 24B*GAL4* and the UAS-GCaMP6 constructs were used in the indicated backgrounds. Imaging was done with a spectral confocal microscope Leica TCS SP. The images were acquired for the times specified in figure legend over 50–75 μm from st. 15 embryos; Z-projections and videos were assembled using Fiji (Schindelin et al, 2012).

## Quantification and statistics

To verify the different distribution of a given marker in wild type and mutant embryos we perform an analysis of several cases with good staining and in the right orientation (n and details provided in the corresponding figure legends). Measurements were imported and treated in Microsoft Excel 16.78.3 where data processing, Statistical analysis and graphical representations were performed. The Mann-Whitney U test was used to determine significant differences between two groups to correct for unequal sample distribution variance.

## Data availability

This study includes no data deposited in external repositories. The source data of this paper are collected in the following database record: https://www.ebi.ac.uk/biostudies/studies/S-BSST1719.

The source data of this paper are collected in the following database record: biostudies:S-SCDT-10_1038-S44319-025-00411-x.

## Peer review information

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

## Acknowledgements

We are grateful to members of our laboratory for helpful discussions, to M. Furriols, P. Giannios and A. Mineo for comments on the manuscript, to B. Hernández for the initial experiments of the project, to A. Letizia for help with the time lapse imaging and to N. Martín, N. Plana and the personnel of the Advanced Digital Microscopy of the IRB Barcelona for technical assistance. Thanks to D. St Johnston and to A. Daga for sharing fly stocks and antibodies. We also thank the Bloomington *Drosophila* Stock Center for providing us fly stocks. This project was supported by grant PID2021-123392NB-I00 funded by MCIN/AEI/10.13039/501100011033 and the European Union and by the Departement d'Universitats i Recerca de la Generalitat de Catalunya through BIST to JC. FT was the recipient of an action from the Erasmus Programme.

## Author contributions

**Delia Ricolo**: Conceptualization; Formal analysis; Validation; Investigation; Visualization; Writing—review and editing. **Francesca Tamba**: Investigation. **Jordi Casanova**: Conceptualization; Formal analysis; Supervision; Funding acquisition; Validation; Investigation; Visualization; Writing—original draft; Project administration; Writing—review and editing.

Source data underlying figure panels in this paper may have individual authorship assigned. Where available, figure panel/source data authorship is listed in the following database record: biostudies:S-SCDT-10_1038-S44319-025-00411-x.

## Disclosure and competing interests statement

The authors declare no competing interests.

# Expanded View Figures

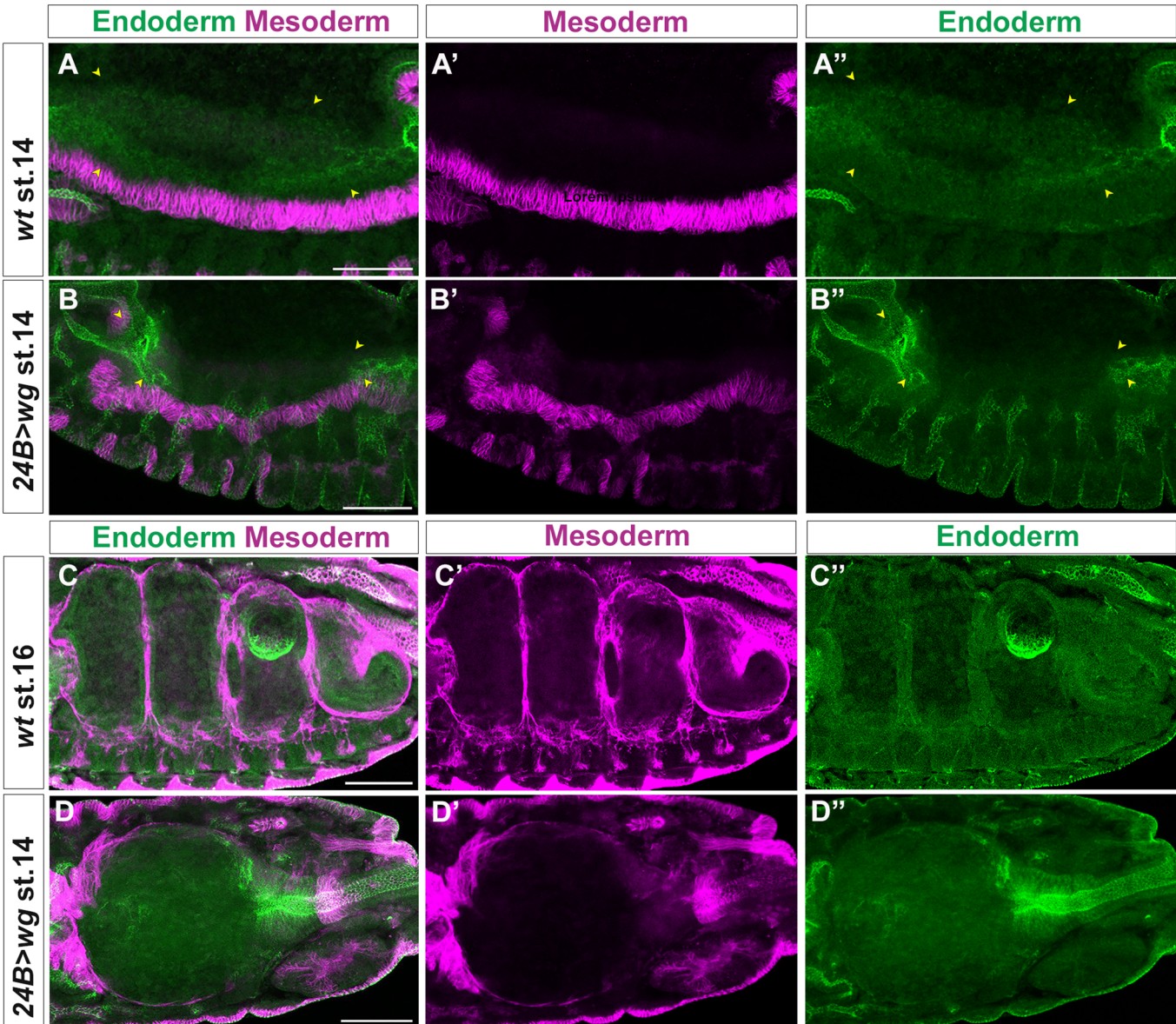

**Figure EV1. Overexpression of wingless in the visceral mesoderm, hinders endoderm migration.**

(A–D) Midgut of wild-type (**A**, **C**) and 24B>*wg* (**B**, **D**) embryos at stage 14th (**A**, **B**) and stage 16th, (**C**, **D**) stained with antibodies against FasIII in magenta and against DECadherin in green. Upon *wg* overexpression (**B**), the row of mesodermal cell does not properly form and the endodermal tissue does not migrate correctly (arrowheads). At later stages, *wg* overexpressing midguts (**D**) completely lose their organization. Anterior is left, dorsal is top, scale bar 50 µm.

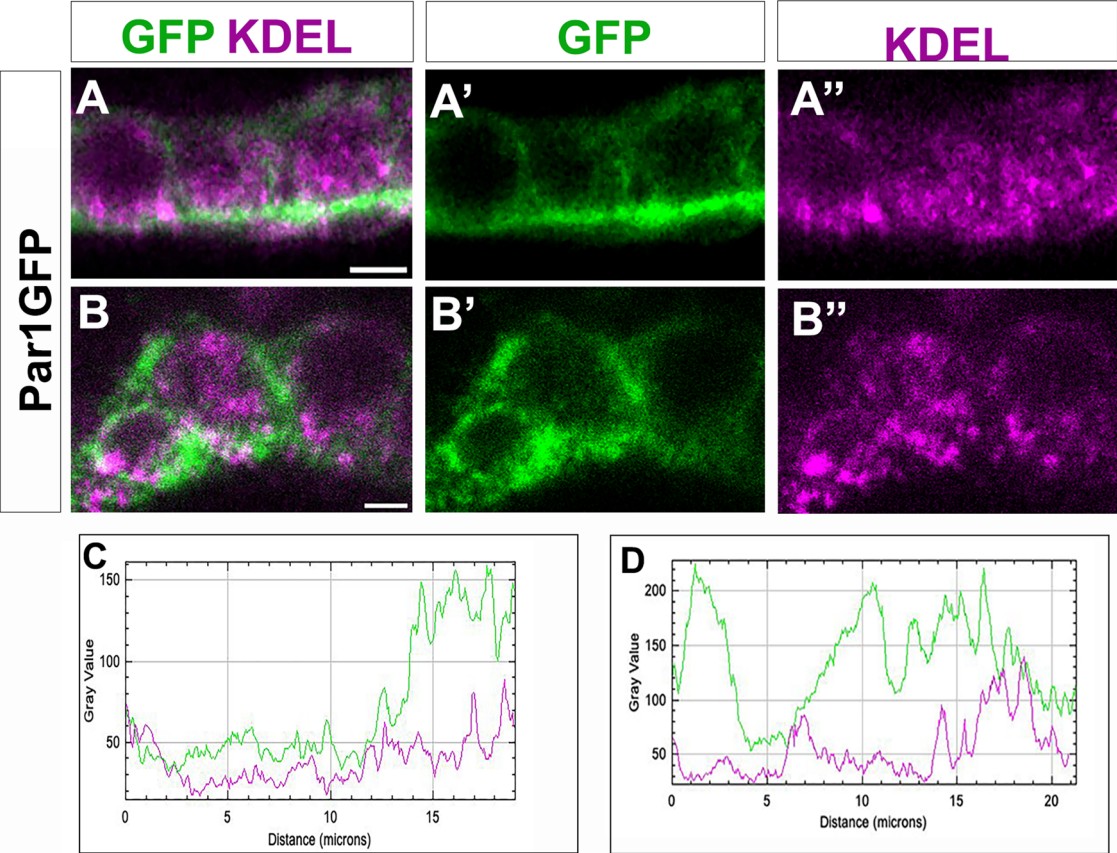

**Figure EV2. Non-uniform localization of Par-1.**

(A, B) Non-constricting (A) and constricting (B) visceral mesodermal cells of a *par1-GFP* embryo stained with GFP (green) and KDEL (magenta) to visualize the ER in the cytoplasm. Internal membrane is up, Scale bar 2 μm. (C, D) Plots showing the differential distribution of Par-1 (green line) and KDEL(magenta line) intensity in visceral mesoderm cells away from the constriction (C) and in constricting visceral mesodermal cells (D).

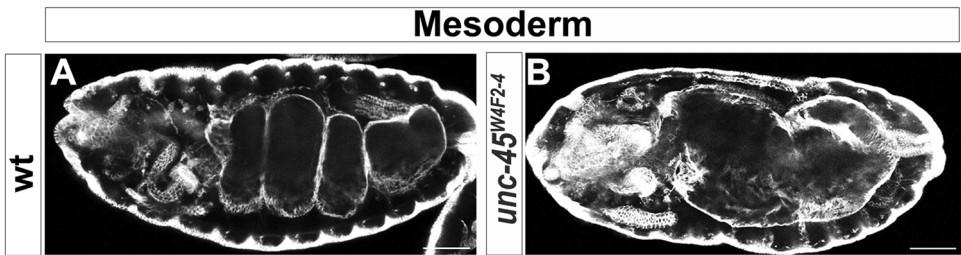

**Figure EV3.** ***unc-45*** **mutant embryos show a midgut constriction failure.**

Wt (**A**) and *unc-45*<sup>4F2-4</sup>(**B**) embryos at stage 16th labelled with antibodies against FasIII to visualise the visceral mesoderm. Embryos are shown in a lateral view, anterior to the left, scale bar 50 μm.

