## [Peer Review File · EMBO Reports]

Autocrine Wingless constricts *Drosophila* embryonic gut by Ca²⁺-mediated repolarisation of mesoderm cells

Delia Ricolo, Francesca Tamba, and Jordi Casanova

Corresponding author(s): Jordi Casanova (jrbmc@ibmb.csic.es) , Delia Ricolo (delia.ricolo@irbbarcelona.org)

Review Timeline:

Submission Date:	14th Jun 24
Editorial Decision:	22nd Jul 24
Revision Received:	18th Nov 24
Editorial Decision:	23rd Jan 25
Revision Received:	29th Jan 25
Accepted:	18th Feb 25

Transaction Report:

Dear Jordi,

As discussed, I took a closer look at the two referee reports we have meanwhile received; the third referee has not yet submitted his/her report.

Referee 2 feels that the relevance of Par-1 and MTs downstream of calcium pulses needs to be tested. Referee 3 raises similar concerns regarding the evidence in support of MT and Par-1 polarization. Both referees consider it likely that the calcium pulses activate myosin rather than organize MTs or activate muscle contractility. These concerns align with my own, when I noted that Cdc-42 dependent Ca^{2+} pulses and Par-1/MTs have not been causally linked. Are Ca^{2+} pulses required for Par-1 polarization dependent on Wg? Could you interfere with Myo2 activity more directly, in addition to the evidence you have from the myosin chaperone.

As both referees feel that the manuscript is interesting and recommend that you should be given a chance to revise your manuscript, I would like to ask you to begin revising your manuscript according to the referees' comments and including the experiments we had discussed earlier by e-mail. Please note that this is a preliminary decision made in the interest of time, and that it is subject to change should the third referee offer very strong and convincing reasons for this. As soon as the third report is in, I will forward it to you.

Please address all referee concerns in a complete point-by-point response. Acceptance of the manuscript will depend on a positive outcome of a second round of review. It is EMBO Reports policy to allow a single round of revision only and acceptance or rejection of the manuscript will therefore depend on the completeness of your responses included in the next, final version of the manuscript.

We realize that it is difficult to revise to a specific deadline. In the interest of protecting the conceptual advance provided by the work, we recommend a revision within 3 months (October 22nd). Please discuss the revision progress ahead of this time with the editor if you require more time to complete the revisions.

I am also happy to discuss the revision further via e-mail or a video call, if you wish.

*****IMPORTANT NOTE:

We perform an initial quality control of all revised manuscripts before re-review. Your manuscript will FAIL this control and the handling will be delayed IN CASE the following APPLIES:

- 1) A data availability section providing access to data deposited in public databases is missing. If you have not deposited any data, please add a sentence to the data availability section that explains that.
- 2) Your manuscript contains statistics and error bars based on $n=2$. Please use scatter blots in these cases. No statistics should be calculated if $n=2$.

When submitting your revised manuscript, please carefully review the instructions that follow below. Failure to include requested items will delay the evaluation of your revision. *****

2) individual production quality figure files as .eps, .tif, .jpg (one file per figure).

Please download our Figure Preparation Guidelines (figure preparation pdf) from our Author Guidelines pages <https://www.embopress.org/page/journal/14693178/authorguide> for more info on how to prepare your figures.

3) a .docx formatted letter INCLUDING the reviewers' reports and your detailed point-by-point responses to their comments. As part of the EMBO Press transparent editorial process, the point-by-point response is part of the Review Process File (RPF),

which will be published alongside your paper.

- 4) a complete author checklist, which you can download from our author guidelines (<<https://www.embopress.org/page/journal/14693178/authorguide>>). Please insert information in the checklist that is also reflected in the manuscript. The completed author checklist will also be part of the RPF.
- 5) Please note that all corresponding authors are required to supply an ORCID ID for their name upon submission of a revised manuscript (<<https://orcid.org/>>). Please find instructions on how to link your ORCID ID to your account in our manuscript tracking system in our Author guidelines (<<https://www.embopress.org/page/journal/14693178/authorguide#authorshipguidelines>>)
- 6) We replaced Supplementary Information with Expanded View (EV) Figures and Tables that are collapsible/expandable online. A maximum of 5 EV Figures can be typeset. EV Figures should be cited as "Figure EV1, Figure EV2" etc... in the text and their respective legends should be included in the main text after the legends of regular figures.
 - For the figures that you do NOT wish to display as Expanded View figures, they should be bundled together with their legends in a single PDF file called *Appendix*, which should start with a short Table of Content. Appendix figures should be referred to in the main text as: "Appendix Figure S1, Appendix Figure S2" etc. See detailed instructions regarding expanded view here: <<https://www.embopress.org/page/journal/14693178/authorguide#expandedview>>
 - Additional Tables/Datasets should be labeled and referred to as Table EV1, Dataset EV1, etc. Legends have to be provided in a separate tab in case of .xls files. Alternatively, the legend can be supplied as a separate text file (README) and zipped together with the Table/Dataset file.
- 7) Please note that a Data Availability section at the end of Materials and Methods is now mandatory. In case you have no data that requires deposition in a public database, please state so instead of refereeing to the database. See also < <https://www.embopress.org/page/journal/14693178/authorguide#dataavailability>>. Please note that the Data Availability Section is restricted to new primary data that are part of this study.
- 8) At EMBO Press we ask authors to provide source data for the main figures. Our source data coordinator will contact you to discuss which figure panels we would need source data for and will also provide you with helpful tips on how to upload and organize the files.

Additional information on source data and instruction on how to label the files are available <<https://www.embopress.org/page/journal/14693178/authorguide#sourcedata>>.

- 9) The journal requires a statement specifying whether or not authors have competing interests (defined as all potential or actual interests that could be perceived to influence the presentation or interpretation of an article). In case of competing interests, this must be specified in your disclosure statement. Further information: <https://www.embopress.org/competing-interests>
- 10) Figure legends and data quantification:
The following points must be specified in each figure legend:
 - the name of the statistical test used to generate error bars and P values,
 - the number (n) of independent experiments (please specify technical or biological replicates) underlying each data point,
 - the nature of the bars and error bars (s.d., s.e.m.)
 - If the data are obtained from n {less than or equal to} 5, show the individual data points in addition to the SD or SEM.
 - If the data are obtained from n {less than or equal to} 2, use scatter blots showing the individual data points.Discussion of statistical methodology can be reported in the materials and methods section, but figure legends should contain a basic description of n, P and the test applied.
See also the guidelines for figure legend preparation:
<https://www.embopress.org/page/journal/14693178/authorguide#figureformat>
 - Please also include scale bars in all microscopy images and define their size in the legend, not the image.

- 11) Our journal encourages inclusion of *data citations in the reference list* to directly cite datasets that were re-used and obtained from public databases. Data citations in the article text are distinct from normal bibliographical citations and should directly link to the database records from which the data can be accessed. In the main text, data citations are formatted as follows: "Data ref: Smith et al, 2001" or "Data ref: NCBI Sequence Read Archive PRJNA342805, 2017". In the Reference list, data citations must be labeled with "[DATASET]". A data reference must provide the database name, accession number/identifiers and a resolvable link to the landing page from which the data can be accessed at the end of the reference.

Further instructions are available at <<https://www.embopress.org/page/journal/14693178/authorguide#referencesformat>>.

12) All Materials and Methods need to be described in the main text using our 'Structured Methods' format, which is required for all research articles. According to this format, the Methods section includes a Reagents and Tools Table (listing key reagents, experimental models, software and relevant equipment and including their sources and relevant identifiers) followed by a Methods and Protocols section describing the methods using a step-by-step protocol format. The aim is to facilitate adoption of the methodologies across labs. More information on how to adhere to this format as well as a downloadable template (.docx) for the Reagents and Tools Table can be found in our author guidelines:

13) As part of the EMBO publication's Transparent Editorial Process, EMBO Reports publishes online a Review Process File to accompany accepted manuscripts. This File will be published in conjunction with your paper and will include the referee reports, your point-by-point response and all pertinent correspondence relating to the manuscript.

Kind regards,

Martina

Referee #2:

In this manuscript, Ricolo et al demonstrate that Wg/Wnt signalling induces the middle constriction of the fly embryonic midgut by activating the transcription factor, Teashirt, which in turn induces expression of the Chloride channel, Clc- α , leading to Ca²⁺ pulses in the constricting mesodermal cells. The authors also present evidence that the constriction is driven by microtubules and propose that myosin activation recruits Par-1 to the endodermal face of the mesoderm cells to drive this process.

The paper is well-written and covers a topic that is likely to interest readers of Embo reports. Their findings identify Teashirt, Clc-a and Calcium signalling as the missing links between the Wg/Wnt pathway and mesoderm constriction. The main weakness lies in the analysis of the roles of Par-1 and microtubules downstream of the Calcium pulses. For a paper proposing a microtubule-driven constriction mechanism, it is surprising that there are no figures showing the organisation of microtubules in the constricting cells versus the non-constricting cells and the EB1 stainings in fixed material do not show any obvious differences. In any case, it is not clear what EB1 labels in fixed samples, as it binds specifically to the plus end of growing microtubules and microtubules do not grow after fixation. I know that imaging the developing visceral mesoderm is challenging, given its localisation in the interior of the embryo, but this leaves a major gap in the logical argument. Given that myosin is thought to be activated by the Calcium pulses, wouldn't a simpler model be that actomyosin contractions provide the force to drive the constrictions and that microtubules play more of a scaffolding function. This seems to me to be the major weakness of the current manuscript and the authors should either exclude the data on microtubules or provide more evidence to support their model.

Possible additional experiments might be:

- 1) Show that Par-1 is specifically required in the constricting cells using WgGal4 or Clc-a-Gal4 to drive Par-1 RNAi
- 2) Demonstrate that the requirement for microtubules and Par-1 is downstream of the Calcium pulses and myosin activation.

3) Provide high resolution images of the microtubules in constricting versus non-constricting cells.

To correlate the Ca²⁺ flux induced MRLC/Sqh activity, Par-1 re-localisation to the membrane cortex facing the endoderm and the microtubule bundling in the constricting mesodermal cells not only in the middle constriction but at least also in the anterior constriction later on, several points need to be addressed:

1. The importance of Par-1 in the middle constricting cells by Wg>Par1-RNAi to show the failure of middle constriction;
2. CIC- α -Gal4>Par1-RNAi to show the failure of both anterior and middle constriction;
3. Par-1's localisation in Tsh mutant and CIC- α mutant mesodermal cells.
4. Ideally, to demonstrate the Par-1's relocalisation is sufficient to trigger the MT bundling and further constriction, one can ectopically express or induce the Par-1's recruitment to the Mys site to where the constriction normally not happening via optogenetics or GFP nanobody recruitment. This won't be a trivial experiment therefore it's not absolutely required for the paper revision.

Referee #3:

This paper from Ricolo and colleagues focuses on the interface between pattern and morphogenesis. The work presented makes substantial progress on understanding the pathway from the homeotic genes to organ morphogenesis, using the particular model of the formation of precise constrictions in the developing midgut of the *Drosophila* embryo. Several steps of the pathway are determined, giving rise to a speculative model of this morphogenetic event. The genetics and phenotypic analysis are strong, the cell biology less so, such that some of the findings are overinterpreted.

I found it particularly interesting to discover that some aspects of the mechanism were shared between the three constrictions and others more specific. So for the Anterior, Middle and Posterior constrictions, all three were dependent on microtubules, Par1, EB1, and Unc45, whereas only the Anterior and Middle were dependent on Tsh and CIC-a, and only the middle on Wg. However, having said this, I had to take the authors word for this in some cases as in the figures it wasn't always easy to see the constriction failures, as some constrictions seemed to be present (e.g. Fig.2 B, which seems to have Anterior and Posterior constrictions). In the calcium imaging movie, it was not easy to see the morphology of the developing gut, and it had three stripes of bright flashes, rather than the expected two. Perhaps some arrows to indicate where the failed constrictions should have formed would make it easier for the readers to appreciate these phenotypes, and marking the calcium flashes as Ant, Mid, would help. Nonetheless I thought sufficient evidence was presented to support the view that all of these molecules contributed in some way to the process of constrictions. I also thought the authors very clearly showed that wingless was acting in an autocrine fashion.

What I did not find convincing was the evidence in support of microtubule polarisation shown in Figure 2 H through O. To me most of the staining of Par1 and EB1 looks simply cytoplasmic/non-nuclear. In H and L the cytoplasm between the nuclei and the internal interface highlighted by betaPS staining is clearly very thin, whereas in the area of constriction, it is clearly larger, and so the difference in Par1 distribution could simply be following the difference in cytoplasmic volume. I did not see a great difference between H' and J' to indicate that Wg altered the distribution of Par1, nor L" and N" to indicate Par1 was altering EB1 polarity. Furthermore, in the end, no model of how polarized microtubules would initiate the constrictions was offered. I did not find the summary figure particularly helpful, as the orientation was rotated relative to all the figures, and how Par1 would impact on the growing ends of microtubules marked with EB1 was not clear nor how microtubules would lead to contraction. I could not see why the authors ruled out a more likely model that the microtubules in some way impact on myosin contractility. One further point that was not clear was what the evidence is that it is non-muscle myosin rather than muscle myosin causing the contraction that leads to the constrictions. As the visceral mesoderm cells are in the process of becoming contractile muscles, perhaps the calcium flash simply induces muscle contraction.

Given the lack of very clear polarity of Par1 in the visceral mesoderm, the final paragraph in the discussion is not warranted. I would prefer to see some discussion on why this may have arisen as a multi-step process rather than Ubx just regulating CIC-a directly, and also whether the timing fits with the multistep process, with wingless synthesis followed by Tsh synthesis followed by CIC-a synthesis. Could the multistep process provide an important delay between pattern and morphogenesis? On this point, some discussion of how synthesising a chloride channel would lead to a very brief calciums flash would be helpful.

In summary, the paper presents numerous solid findings, but either they need to either strengthen the evidence for polarized distribution of Par1 and EB1 (e.g. double label with a cytoplasmic marker to show that its not the same) or reduce the emphasis on this aspect.

Referee 2 feels that the relevance of Par-1 and MTs downstream of calcium pulses needs to be tested. Referee 3 raises similar concerns regarding the evidence in support of MT and Par-1 polarization. Both referees consider it likely that the calcium pulses activate myosin rather than organize MTs or activate muscle contractility. These concerns align with my own, when I noted that Clc-a dependent Ca²⁺ pulses and Par-1/MTs have not been causally linked. Are Ca²⁺ pulses required for Par-1 polarization dependent on Wg?

In the original version of the manuscript, we showed that EB1 and Par-1 distribution in the constricted cells depends on the Clc-a chloride channel, which is required for the calcium fluxes. Now, following the suggestion by both referee 2 and yourself, we have analysed EB1 distribution upon inactivation of the SERCA calcium pump, which removes midgut folds (Mitchell et al. 2022) and confirmed EB1 distribution to be downstream of the calcium pulses. Likewise, following the suggestion from referee 3, we have used a cytoplasmic marker to check whether the distribution of Par-1 was just a consequence of the cytoplasm redistribution and found that this is not the case as Par1 is enriched in a particular area of the cytoplasm.

Could you interfere with Myo2 activity more directly, in addition to the evidence you have from the myosin chaperone.

Your comments and those from the referees' evidence that we have not succeeded in exposing adequately the previous work from other authors that are essential for our present work. In particular, the work by Mitchell et al have already shown that they can interfere with the gut folding by RNA interference of the gene encoding the myosin light chain kinase. This is one of the reasons we did not reproduce the experiment but, alternatively, aimed at triggering the Unc-45 Myosin chaperone, which in addition has been directly linked to Par-1 polarisation (Doerflinger et al 2022). We have modified the text accordingly.

Below we address the specific points raised by both referees.

Referee #2:

In this manuscript, Ricolo et al demonstrate that Wg/Wnt signalling induces the middle constriction of the fly embryonic midgut by activating the transcription factor, Teashirt, which in turn induces expression of the Chloride channel, Clc- α , leading to Ca²⁺ pulses in the constricting mesodermal cells. The authors also present evidence that the constriction is driven by microtubules and propose that myosin activation recruits Par-1 to the endodermal face of the mesoderm cells to drive this process.

The paper is well-written and covers a topic that is likely to interest readers of Embo reports. Their findings identify Teashirt, Clc-a and Calcium signalling as the missing links between the Wg/Wnt pathway and mesoderm constriction. The main weakness lies in the analysis of the roles of Par-1 and microtubules downstream of the Calcium pulses.

In the original version of the manuscript, we showed that EB1 and Par-1 distribution in the constricted cells depends on the ClC-a chloride channel, which is required for the calcium fluxes. Now, following the suggestion by both referee 2 and yourself, we have analysed EB1 distribution upon inactivation of the SERCA calcium pump, which removes midgut folds (Mitchell et al. 2022) and confirmed EB1 relocalisation to be downstream of the calcium pulses.

For a paper proposing a microtubule-driven constriction mechanism, it is surprising that there are no figures showing the organisation of microtubules in the constricting cells versus the non-constricting cells and the EB1 stainings in fixed material do not show any obvious differences.

Previous work from Reuter and Scott (1990) already showed by electronic microscopy that the constricting cells display a transient and dense accumulation of microtubule bundles oriented in the direction of the constriction, a microtubule arrangement that coincides with the time of constriction as it is not detected at earlier or later times. As this point was already very clear we did not intend to show microtubule organization by confocal microscopy, which would be hardly as informative as the electronic microscopy pictures already published. However, following the referee's suggestion, in the new version we address the reader directly to the published EM picture from the Reuter and Scott paper.

In any case, it is not clear what EB1 labels in fixed samples, as it binds specifically to the plus end of growing microtubules and microtubules do not grow after fixation.

We apologise because there was not enough detail regarding our fixing protocols. In this and previous work we needed to detect both microtubules and other cell components within the whole embryo and have tested many different conditions such as methanol fixation, cold fixation, boiling fixation and the t10 min fixing protocol; the latter is the one that gives us the best results for all structures we need to detect and is the one we use in this work (see the discussion on the peer review section of Ricolo and Araujo 2020). We clarify now this issue in the material and methods section of the revised manuscript. We also agree about the fixation effect on microtubule growth but we think this does not prevent using this methodology to have a snapshot of EB1 accumulation at the time of fixation.

I know that imaging the developing visceral mesoderm is challenging, given its localisation in the interior of the embryo, but this leaves a major gap in the logical argument. Given that myosin is thought to be activated by the Calcium pulses, wouldn't a simpler model be that actomyosin contractions provide the force to drive the constrictions and that microtubules play more of a scaffolding function. This seems to me to be the major weakness of the current manuscript and the authors should either exclude the data on microtubules or provide more evidence to support their model.

We agree with the possibility suggested by the referee and indeed we find it is not incompatible with the results reported. While our work indicates a role of myosin in

cell polarisation and microtubule organization and shows that both are required for the middle gut constriction, calcium activated myosin might also contribute to actomyosin contractions providing the force to drive the constrictions. Alternatively, recent work has underlined the role of microtubule forces in cell reshaping. We now clarify and discuss this issue in the revised version.

Possible additional experiments might be:

1) Show that Par-1 is specifically required in the constricting cells using WgGal4 or Clc-a-Gal4 to drive Par-1 RNAi

We have done the experiment suggested by driving Par-1 RNAi in the constricting cells with the wgGal4 construct. We include the new data in the revised version of the manuscript and add an image as a new panel in the present figure 2

2) Demonstrate that the requirement for microtubules and Par-1 is downstream of the Calcium pulses and myosin activation.

The conservation of the elements involved in the polarisation of Par-1 as well as their common phenotypes pointed to us to the conservation of the mechanism required for polarisation. However, as indicated above and following the suggestion, we have analysed EB1 distribution upon inactivation of the SERCA calcium pump, which removes midgut folds (Mitchell et al. 2022) and confirmed EB1 relocalisation to be downstream of the calcium pulses. We have added this information to the current version of the manuscript.

3) Provide high resolution images of the microtubules in constricting versus non-constricting cells.

Please, see above

To correlate the Ca²⁺ flux induced MRLC/Sqh activity, Par-1 re-localisation to the membrane cortex facing the endoderm and the microtubule bundling in the constricting mesodermal cells not only in the middle constriction but at least also in the anterior constriction later on, several points need to be addressed:

1. The importance of Par-1 in the middle constricting cells by Wg>Par1-RNAi to show the failure of middle constriction;

Please, see above

2. CIC- α -Gal4>Par1-RNAi to show the failure of both anterior and middle constriction;

We agree this could be a good experiment to show the requirement of Par-1 for both anterior and middle constriction but we think that the above requested and performed experiment (wg>Par1-RNAi) as well as the already present in the first version of the manuscript (24B>Par1-RNAi) clearly answers the issue raised by the referee.

3. Par-1's localisation in Tsh mutant and CIC- α mutant mesodermal cells.

We are sorry but, while we can observe the absence of the gut constriction in the *CIC-a* mutants, the mesoderm is too disorganised to properly assess its intracellular Par-1 distribution. Therefore, we have approached the referee request by an alternative experiment showing that EB1 relocalisation is affected by the calcium flux, which we also show to be affected in *CIC-a* mutants

4. Ideally, to demonstrate the Par-1's relocalisation is sufficient to trigger the MT bundling and further constriction, one can ectopically express or induce the Par-1's recruitment to the Mys site to where the constriction normally not happening via optogenetics or GFP nanobody recruitment. This won't be a trivial experiment therefore it's not absolutely required for the paper revision.

As mentioned by the referee, this is not a trivial experiment. Indeed, these are very tiny cells, at the interior of the embryo, contracting for a very short period of time. Indeed, many of the provided results were already quite challenging (i.e., calcium pulses in the gut constrictions of wildtype and mutant embryos). Alternatively, we attempted a similar result as the one suggested by the referee by ectopically expressing *wg* all over the visceral mesoderm with the hope to force a Par-1 relocalisation. Unfortunately, whether this caused a change in Par-1 localisation, at the same time inhibited the visceral mesoderm migration with did not allow us to check for an extra constriction phenotype

Referee #3:

This paper from Ricolo and colleagues focuses on the interface between pattern and morphogenesis. The work presented makes substantial progress on understanding the pathway from the homeotic genes to organ morphogenesis, using the particular model of the formation of precise constrictions in the developing midgut of the *Drosophila* embryo. Several steps of the pathway are determined, giving rise to a speculative model of this morphogenetic event. The genetics and phenotypic analysis are strong, the cell biology less so, such that some of the findings are overinterpreted.

I found it particularly interesting to discover that some aspects of the mechanism were shared between the three constrictions and others more specific. So for the Anterior, Middle and Posterior constrictions, all three were dependent on microtubules, Par1, EB1, and Unc45, whereas only the Anterior and Middle were dependent on Tsh and *CIC-a*, and only the middle on *Wg*. However, having said this, I had to take the authors word for this in some cases as in the figures it wasnt always easy to see the constriction failures, as some constrictions seemed to be present (e.g. Fig.2 B, which seems to have Anterior and Posterior constrictions).

Yes, the referee is right, the embryo in Fig.2B has anterior and posterior constriction being the middle one the only not developing. This is however what would be expected in this particular embryo, as the UAS-Spastin construct is driven by *wg-gal4* which is expressed in the middle constriction but not in the anterior and posterior ones. We are

sorry if the figure confused the referee as this embryo was the only one in the column in which we had specifically triggered the middle constriction. We have tried to make clearer this point in the new figure legend.

In the calcium imaging movie, it was not easy to see the morphology of the developing gut, and it had three stripes of bright flashes, rather than the expected two. Perhaps some arrows to indicate where the failed constrictions should have formed would make it easier for the readers to appreciate these phenotypes, and marking the calcium flashes as Ant, Mid, would help.

These are challenging images to have, as we are focusing in a small group of cells, at the interior of the whole living embryo and that are contracting for a short period of time. We have followed the suggestions of the referee in the new version and we hope now it is easier to see.

What I did not find convincing was the evidence in support of microtubule polarisation shown in Figure 2 H through O. To me most of the staining of Par1 and EB1 looks simply cytoplasmic/non-nuclear. In H and L the cytoplasm between the nuclei and the internal interface highlighted by betaPS staining is clearly very thin, whereas in the area of constriction, it is clearly larger, and so the difference in Par1 distribution could simply be following the difference in cytoplasmic volume. I did not see a great difference between H' and J' to indicate that wg altered the distribution of Par1, nor L" and N" to indicate Par1 was altering EB1 polarity.

As mentioned above, some of these pictures are not easy to obtain and a clear conclusion only comes by the thoughtful analysis of several cases with good staining and in the right orientation. Therefore, we have performed this analysis and quantified the results accordingly. We have made explicit the procedure in the new version. Nevertheless, the issue raised by the referee is very relevant, even more as Par-1 has been described not as a membrane protein but instead as a protein accumulated cortically (we have clarified this in the new version). Thus, according to his suggestion, we have done a double staining to examine at the same time Par-1 distribution and a cytoplasmic protein. This experiment allows us to assess that Par-1 is not uniformly distributed in the cytoplasm but instead shows an asymmetric distribution, which is modified upon wg signalling.

Furthermore, in the end, no model of how polarized microtubules would initiate the constrictions was offered. I did not find the summary figure particularly helpful, as the orientation was rotated relative to all the figures, and how Par1 would impact on the growing ends of microtubules marked with EB1 was not clear nor how microtubules would lead to contraction.

Yes, our work points to the mechanism and the role of cell polarisation and microtubule organization for the middle gut constriction. Par-1 has already been shown to be able to regulate microtubules (i.e., Doerflinger et al., 2003), which provides a link between the two elements and different alternatives have been proposed for the role of microtubules in cell reshaping (i.e., see the review by Röper 2020). We now clarify

and discuss this issue in the revised version. We have also modified the summary figure

I could not see why the authors ruled out a more likely model that the microtubules in some way impact on myosin contractility. One further point that was not clear was what the evidence is that it is non-muscle myosin rather than muscle myosin causing the contraction that leads to the constrictions. As the visceral mesoderm cells are in the process of becoming contractile muscles, perhaps the calcium flash simply induces muscle contraction.

We agree with the possibility suggested by the referee and indeed we find it is not incompatible with the results reported. While our work indicates a role of myosin in cell polarisation and microtubule organization and show that both are required for the middle gut constriction, calcium activated myosin might also contribute to actomyosin contractions providing the force to drive the constrictions. Alternatively, recent work has underlined the role of microtubule forces in cell reshaping. We now clarify and discuss this issue in the revised version (see the point above).

Given the lack of very clear polarity of Par1 in the visceral mesoderm, the final paragraph in the discussion is not warranted.

We have rephrased the final paragraph in the discussion according to the new results on Par1 from the experiments suggested by the referee as well on the polar distribution of integrin at the cell membrane.

I would prefer to see some discussion on why this may have arisen as a multi-step process rather than Ubx just regulating *CIC-a* directly, and also whether the timing fits with the multistep process, with wingless synthesis followed by Tsh synthesis followed by *CIC-a* synthesis. Could the multistep process provide an important delay between pattern and morphogenesis? On this point, some discussion of how synthesising a chloride channel would lead to a very brief calciums flash would be helpful.

We have extended the discussion to encompass the issues raised by the referee. Thus, for instance, we think an advantage of the current model of gut constriction regulation is that it allows a constriction to be generated at a precise position and a specific position, which would be difficult if Ubx had to regulate directly *CIC-a* expression due to the broad and persistent domain of expression of the former.

In summary, the paper presents numerous solid findings, but either they need to either strengthen the evidence for polarized distribution of Par1 and EB1 (e.g. double label with a cytoplasmic marker to show that its not the same) or reduce the emphasis on this aspect.

We hope the referee will consider that we have now a good balance with the new results about EB1 and Par-1 distribution and the rewritten results and discussion of the current manuscript.

Dear Jordi,

Thank you for the submission of your revised manuscript to EMBO reports. I apologize again for the delay in handling your manuscript, which is in part due to the Christmas/New Year period and to the fact that I have discussed the referee reports and remaining concerns with you and two expert advisors.

As you know, referee #2 raised concerns regarding the conclusiveness of the data on Par-1 and MT polarity downstream of Ca²⁺ spikes. The referee suggested to remove these data and focus on the autocrine effect of Wg. I have discussed the remaining concerns and the data at hand with two additional independent experts in the field.

Both advisors considered your revision overall adequate and supported publication at EMBO Reports. I copy the advisors comments below my signature. I note however that the advisors agreed in part with the concerns raised by referee #2, regarding causality of Par-1 and MT polarity in mediating constriction and the concern that fixation can affect MT dynamics. Please discuss these limitations and concerns in the manuscript and please also address all referee concerns in a point-by-point response.

Browsing through the manuscript myself, I noticed a few editorial things that we need before we can proceed with the official acceptance of your study.

- Figure 5 appears to have rather low resolution. The protein labels appear blurred.
 - Please update the 'Conflict of interest' paragraph to our new 'Disclosure and competing interests statement'. For more information see <https://www.embopress.org/page/journal/14693178/authorguide#conflictsofinterest>
 - References: the abbreviation 'et al' should be used if there are more than 10 authors, i.e., the first 10 authors are listed followed by 'et al'. You can download the respective EndNote file from our Guide to Authors https://endnote.com/style_download/embo-reports/
 - Please provide a complete author checklist, which you can download from our author guidelines (<<https://www.embopress.org/page/journal/14693178/authorguide>>). Please insert information in the checklist that is also reflected in the manuscript. The completed author checklist will also be part of the RPF.
 - Please provide Figure EV2 and EV3 in either .eps, .tif, or .jpg format instead of .psd.
 - Please add callouts in the text for Fig. 2A, Fig. 3E, Fig. 4A, Fig. 4C, Fig. 5
 - Movies: the nomenclature (source file names, title, legends, manuscript callouts) should be corrected to Movie EV1, Movie EV2; each legend should be provided in a readme.txt file and then should be zipped together with its movie file so that we have two zip folders - Movie EV1 and Movie EV2.
 - Materials and Methods should be Methods.
 - When submitting your revised manuscript, please remove the Reagents and Tools Table in the Methods section of the manuscript and upload it as a separate file choosing the file type "Reagent Table".
An example of a Method paper with Structured Methods can be found here: <https://www.embopress.org/doi/10.15252/msb.20178071>.
 - Data availability: Please provide a URL that resolves directly to the BioStudies deposition, not just the accession number.
 - During our routine image analysis, which we perform on all revised manuscripts, we noticed that the image of the FasIII stained WT embryo in Figure 3E seems to have been reused in Fig. EV3A. Please check and if this is the case, please clearly indicate the re-use in the respective figure legends. If at hand, a different image would be preferable. The same is true for the WT image shown in Figure 4A, but it might also be the highly uniform nature of the embryonic morphology in this case?
 - Our production/data editors have asked you to clarify several points in the figure legends (see below). Please incorporate these changes in the manuscript and return the revised file with tracked changes with your final manuscript submission.
- A) Replicates and error bars:
- Please define the annotated p values ***/** as well as provide the exact p-values for the same in the legend of figure 2l, p; 4e; as appropriate.
 - Please indicate the statistical test used for data analysis in the legends of figures 2l, p; 4e.
 - Please note that the box plots need to be defined in terms of minima, maxima, centre, bounds of box and whiskers, and

percentile in the legends of figures 2l, p; 4e.

B) Data presentation:

- Please note that the scale bar needs to be defined for figures 2h-k, m-n; 3e-j; 4a-b.
- Please note that scale bar and its definition are missing for figures 3k-l.

- Finally, EMBO Reports papers are accompanied online by

A) a short (1-2 sentences) summary of the findings and their significance,

B) 2-3 bullet points highlighting key results and

C) a schematic summary figure that provides a sketch of the major findings (not a data image).

Please provide the summary figure as a separate file in PNG or JPG format at a size of 550x300-600 pixels (width x height).

Please note that the size is rather small and that text needs to be readable at the final size. Please send us this information along with the revised manuscript.

With kind regards,

Martina

=====

Referee #2:

The revised manuscript from Ricolo et al has improved compared to the previous version and is now more circumspect in its conclusions. Their data convincingly demonstrate that Wingless acts in an autocrine fashion to turn on the expression of Teashirt, which in turn activates the Clc Chloride channel to trigger Calcium spikes in the mesoderm cells that drive the middle invagination in the midgut. These are important observations that tell an interesting story about gut morphogenesis. However, the data suggesting that the Calcium spikes act through Par-1 and microtubule reorganisation are still weak. Although the authors' quantifications seem very significant, their images of GFP-PAR-1 and EB1 in constricting versus non-constricting mesoderm cells remain unconvincing. The only significant improvement was the use of KDEL as a "cytoplasmic marker" to show that GFP-Par-1 is enriched on the non-endoderm facing side of non-constricting cells in Figure EV2. This does show a clear enrichment on Par-1 on one side of the cell, but we are not shown a constricting cell for comparison.

The weakness of these results on Par-1 and EB1 contrast with the quality of the other data in the paper and I think that they should either be strengthened or removed before publication. Whether this manuscript is still suitable for EMBO Reports without these data will be a decision for the editors.

Specific points:

1) I do not think that EB1 can be used as a marker for microtubule plus ends in fixed samples. In fact, its enrichment most likely marks where minus ends are concentrated, since microtubules depolymerise from their plus ends during fixation and EB1 ends up on the plus ends of the more stable stumps of microtubules at the centrosomes or other microtubule organising centres.

2) KDEL is not a cytoplasmic marker but an ER marker.

Referee #3:

The authors have done a good job addressing the points I raised in my review, and the manuscript is now suitable for publication in EMBO reports. I noted a few typos:

p5 visceral blastoderm should be visceral mesoderm

i.e., Doerflinger should be e.g. Doerflinger

there is a truncated sentence starting on the last line as .middle

p7 second paragraph, problem with grammar in second sentence., last word of paragraph is misspelled, as is the last word in the next paragraph

=====

Advisor #1:

I read the paper carefully and find it very clear and definitely appropriate for EMBO Reports. While I agree with reviewer 2 that the precise contribution of Par-1 and MTs to the constriction mechanism is not entirely clear (e. g. whether the force for constriction is mainly generated by actomyosin contractility or by some dynamic movement of Mts), the data reported in this paper regarding this point are valid and important and should remain in the manuscript.

Advisor #2:

My feeling is that the authors have made reasonable revisions. Although fixation can affect MT dynamics (and thereby precise EB1 localisation), the authors show a polarized distribution of EB1 that is altered with par-1 mutants (both with fixation). This is consistent with a functional relationship between MT organization and Par1 and their overall model.

Point-by-point response to both your demands and the concerns raised by the referees.

- Figure 5 appears to have rather low resolution. The protein labels appear blurred.

We now provide a new version of Figure 5

- Please update the 'Conflict of interest' paragraph to our new 'Disclosure and competing interests statement'. For more information see

We have updated the title of the statement

- References: the abbreviation 'et al' should be used if there are more than 10 authors, i.e., the first 10 authors are listed followed by 'et al'. You can download the respective EndNote file from our Guide to Authors

We have modified the references according to the rule

- Please provide a complete author checklist, which you can download from our author guidelines

We provide the complete author checklist

- Please provide Figure EV2 and EV3 in either .eps, .tif, or .jpg format instead of .psd.

We provide these figures in the new format

- Please add callouts in the text for Fig. 2A, Fig. 3E, Fig. 4A, Fig. 4C, Fig. 5

We have added the corresponding callouts in the text

- Movies: the nomenclature (source file names, title, legends, manuscript callouts) should be corrected to Movie EV1, Movie EV2; each legend should be provided in a readme.txt file and then should be zipped together with its movie file so that we have two zip folders - Movie EV1 and Movie EV2.

We provide the two zip folders

- Materials and Methods should be Methods.

We have changed the heading

- When submitting your revised manuscript, please remove the Reagents and Tools Table in the Methods section of the manuscript and upload it as a separate file choosing the file type "Reagent Table".

We provide the new file

- Data availability: Please provide a URL that resolves directly to the BioStudies deposition, not just the accession number.

https://www.ebi.ac.uk/biostudies/submissions/edit/S-BSST1719;method=PAGE_TAB

- During our routine image analysis, which we perform on all revised manuscripts, we noticed that the image of the FasIII stained WT embryo in Figure 3E seems to have been reused in Fig. EV3A. Please check and if this is the case, please clearly indicate

the re-use in the respective figure legends. If at hand, a different image would be preferable.

We have included a different image in Fig. EV3A

The same is true for the WT image shown in Figure 4A, but it might also be the highly uniform nature of the embryonic morphology in this case?

In Figure 4A, the image corresponds to another embryo, a younger one still missing the anterior constriction

- Please define the annotated p values */** as well as provide the exact p-values for the same in the legend of figure 2l, p; 4e; as appropriate.**

We have done as requested

- Please indicate the statistical test used for data analysis in the legends of figures 2l, p; 4e.

We have done as requested

- Please note that the box plots need to be defined in terms of minima, maxima, centre, bounds of box and whiskers, and percentile in the legends of figures 2l, p; 4e.

We have done as requested

- Please note that the scale bar needs to be defined for figures 2h-k, m-n; 3e-j; 4a-b.

We have done as requested

- Please note that scale bar and its definition are missing for figures 3k-l.

These figures, as indicated, are just snapshots from the provided movies, which makes it hard to include appropriate scale bars. May we let them as they are?

- Finally, EMBO Reports papers are accompanied online by

**A) a short (1-2 sentences) summary of the findings and their significance,
B) 2-3 bullet points highlighting key results and**

We have added the short summary and the bullet points in the manuscript

C) a schematic summary figure that provides a sketch of the major findings (not a data image). Please provide the summary figure as a separate file in PNG or JPG format at a size of 550x300-600 pixels (width x height). Please note that the size is rather small and that text needs to be readable at the final size. Please send us this information along with the revised manuscript.

We would like to suggest to use as a schematic summary the image in Figure 5, which in fact is a sketch of the major findings.

Referee #2:

The revised manuscript from Ricolo et al has improved compared to the previous version and is now more circumspect in its conclusions. Their data convincingly demonstrate that Wingless acts in an autocrine fashion to turn on the expression of Teashirt, which in turn activates the Clc Chloride channel to trigger Calcium spikes in the mesoderm cells that drive the middle invagination in the midgut. These are

important observations that tell an interesting story about gut morphogenesis. However, the data suggesting that the Calcium spikes act through Par-1 and microtubule reorganisation are still weak. Although the authors' quantifications seem very significant, their images of GFP-PAR-1 and EB1 in constricting versus non-constricting mesoderm cells remain unconvincing. The only significant improvement was the use of KDEL as a "cytoplasmic marker" to show that GFP-Par-1 is enriched on the non-endoderm facing side of non-constricting cells in Figure EV2. This does show a clear enrichment on Par-1 on one side of the cell, but we are not shown a constricting cell for comparison.

We would like to point out that besides the use of KDEL there are also significant additional data in the new version, in particular the very difficult one about the direct role of Ca²⁺ in cell components polarisation as had been requested. Responding to the criticism by the referee, we now provide the image of KDEL and Par1GFP in a constricting cell for comparison as requested.

Specific points:

1) I do not think that EB1 can be used as a marker for microtubule plus ends in fixed samples. In fact, its enrichment most likely marks where minus ends are concentrated, since microtubules depolymerise from their plus ends during fixation and EB1 ends up on the plus ends of the more stable stumps of microtubules at the centrosomes or other microtubule organising centres.

As we answered in our previous reply, there is a lot of debate about this point and indeed many papers have used EB1 in our same conditions as a plus end microtubule marker. However, even if the referee does not agree on that, he/she states in this point that EB1 is a marker for MT polarity (he/she claims it is a plus ends marker). So, even in the view of this referee, which differs from the other referee, EB1 is still a marker of MT polarity. We have rewritten the text to discuss this issue.

2) KDEL is not a cytoplasmic marker but an ER marker.

Yes, KDEL is an ER marker and we state this on the manuscript. We considered many putative "uniform" cytoplasmic markers and we thought an ER marker was a good marker as the ER is distributed over the cytoplasm. In particular, the use of KDEL allows us to show that there are cytoplasmic regions detected by KDEL that are devoid or very poor in Par1 accumulation and thus that Par1 distribution is not simply following the cytoplasmic volume. We make this point clearer in the present version.

Referee #3:

The authors have done a good job addressing the points I raised in my review, and the manuscript is now suitable for publication in EMBO reports. I noted a few typos:

p5 visceral blastoderm should be visceral mesoderm

i.e., Doerflinger should be e.g. Doerflinger

there is a truncated sentence starting on the last line as .middle

p7 second paragraph, problem with grammar in second sentence., last word of paragraph is misspelled, as is the last word in the next paragraph

We are grateful to the referee for pointing these typos and we have corrected them

Dr. Jordi Casanova
Institut de Biologia Molecular de Barcelona (CSIC), Institute for Research in Biomedicine Barcelona
Cells and Tissues
C/ Baldiri Reixac 10-12
Barcelona, Catalonia 08028
Spain

Dear Jordi,

Thank you very much for submitting your revised manuscript and for implementing all editorial requests. I am very pleased to accept your manuscript for publication in the next available issue of EMBO reports. Thank you for your contribution to our journal.

Kind regards,

Martina
